# Emerging Concepts of Hybrid Epithelial-to-Mesenchymal Transition in Cancer Progression

**DOI:** 10.3390/biom10111561

**Published:** 2020-11-16

**Authors:** Dona Sinha, Priyanka Saha, Anurima Samanta, Anupam Bishayee

**Affiliations:** 1Department of Receptor Biology and Tumour Metastasis, Chittaranjan National Cancer Institute, Kolkata 700 026, India; poojasaha.saha79@gmail.com (P.S.); anurimasamanta@gmail.com (A.S.); 2Lake Erie College of Osteopathic Medicine, Bradenton, FL 34211, USA

**Keywords:** collective migration, epithelial/mesenchymal phenotype, phenotypic stability factor, hybrid/partial EMT, metastasis, stemness

## Abstract

Epithelial mesenchymal transition (EMT) is a complex process through which epithelial (E) cells lose their adherens junctions, transform into mesenchymal (M) cells and attain motility, leading to metastasis at distant organs. Nowadays, the concept of EMT has shifted from a binary phase of interconversion of pure E to M cells and vice versa to a spectrum of E/M transition states preferably coined as hybrid/partial/intermediate EMT. Hybrid EMT, being a plastic transient state, harbours cells which co-express both E and M markers and exhibit high tumourigenic properties, leading to stemness, metastasis, and therapy resistance. Several preclinical and clinical studies provided the evidence of co-existence of E/M phenotypes. Regulators including transcription factors, epigenetic regulators and phenotypic stability factors (PSFs) help in maintaining the hybrid state. Computational and bioinformatics approaches may be excellent for identifying new factors or combinations of regulatory elements that govern the different EMT transition states. Therapeutic intervention against hybrid E/M cells, though few, may evolve as a rational strategy against metastasis and drug resistance. This review has attempted to present the recent advancements on the concept and regulation of the process of hybrid EMT which generates hybrid E/M phenotypes, evidence of intermediate EMT in both preclinical and clinical setup, impact of partial EMT on promoting tumourigenesis, and future strategies which might be adapted to tackle this phenomenon.

## 1. Introduction

Epithelial mesenchymal transition (EMT), is a complicated cellular process in which epithelial (E) cells lose adherence through cellular junctions, get deprived of apico-basolateral polarity, gain migratory potential and are transformed to mesenchymal (M) phenotype. EMT has been frequently associated with embryonic development, wound healing and cancer progression. During embryonic development, embryonic cells undergo EMT in order to migrate to a distant site and subsequently follow the reverse process, M-to-E transition (MET), to differentiate into various cell types. Likewise, cancer cells reactivate the EMT program during delamination from a primary tumour, migrate to distant organs, and form secondary tumours [1]. EMT is accompanied with disruption of adherens junctions and tight junctions, resulting in the dissemination of E cells. EMT aggravates the expression of M marker proteins, promotes attachment to the extracellular matrix (ECM) and acquisition of spindle-shaped M phenotype, enhances the migratory potential of an individual cell by reorganizing actin fibres and invades through basement membranes, blood and lymphatic vessels. Subsequent to intravasation, the cells primed for EMT survive as circulating tumour cells (CTCs) and ultimately extravasate into distant organs [2]. The CTCs harbour both E and M traits and exhibit hybrid E/M phenotypes (Figure 1) which impart them with plasticity and a survival advantage in different microenvironments during metastasis [3]. After extravasation, tumour cells need to undergo MET for metastatic colonization and growth at secondary sites. EMT endows cancer cells with aggressive phenotypes, drug resistance, aids suppression of senescence, anoikis, immune surveillance and acquisition of cancer stem cell (CSC)-like characteristics [4]. Conventional EMT has been often defined by the cadherin (cad) switch which indicates depletion of the epithelial marker E-cadherin (E-cad) and the gain of the M marker neural (N)-cadherin (N-cad). Migration of fibroblast-like M cells is mediated by integrin regulated interaction between cells and ECM [5]. EMT is governed by a multitude of factors which include ECM components, hypoxia, exosomes, non-coding RNAs, and growth factors such as hepatocyte growth factor (HGF), fibroblast growth factor (FGF) and transforming growth factor-β (TGF-β) [4]. The mobility of the tumour cells is greatly influenced by the tumour microenvironment (TME) and the proximity to blood vessels [6]. EMT is frequently coupled with poorly differentiated tumours arising from different tissues.

Recent studies indicated that instead of the binary process of EMT, occurrence of hybrid E/M phenotypes is a gradual transformation of E cells, through a number of intermediate stages to M form [7,8]. Partial/intermediate/hybrid EMT enables cancer cells to undergo migration and ECM attachment by virtue of their co-existing E and M characters, respectively, leading to stemness and tumour progression. There are some excellent reviews on specified aspects, such as CTCs [3]; role of EMT or partial EMT on cancer pathogenesis [9], metastasis [10], stemness, immune profile [11], and therapy resistance [12]. However, in this review, we have attempted to provide a complete knowledge oriented with the recent advancements on the concept and regulation of the process of hybrid EMT which generates hybrid E/M phenotypes, evidences of intermediate EMT in both preclinical and clinical setup, impact of partial EMT on promoting tumourigenesis and future strategies which might be adapted to tackle this phenomenon. Availability of the wide spectrum of partial EMT information on a single platform might be beneficial for both basic and clinical researchers.

## 2. Hybrid E/M and Their Plasticity

Hybrid E/M phenotypes exhibit loss of apical–basal polarity, better motility while conserving adhesive property with neighboring cells and acquire M features [6]. The plasticity of the tumour cells undergoing EMT vary according to their stage. The early hybrid E/M subpopulation shows inclination towards a hybrid E/M phenotype, the M subpopulation towards an M phenotype without spontaneous reversal towards an E phenotype and the hybrid E/M cells exhibit maximum plasticity as they are potent enough to give rise to both E and M phenotypes [13]. EMT or MET are not “all-or-none” processes; rather, cells can achieve one or more partial E/M states that may be more plastic in comparison with fully E or M cells [14]. In other words, the process of partial EMT is a dynamic transition of cells from the E to M phase. The intermediate E/M stages exhibiting different levels of E and M markers, transcriptional, and epigenetic characteristics are specially localized at the invading edge of the tumour [6]. E-cad is depleted at the invading edge of the tumour but rich at the centre of the tumour [4]. Migration of multicellular clusters is exhibited by leading cells with hybrid E/M phenotypes (elevated M characteristics and actin-induced mobility) compared to central cells with E phenotypes (intact polarity and intercellular junctions and migration due to the leader cell mediated traction forces). In case of single-cell migration a full E/M phenotype is a prerequisite which signifies reduction of adhesive property, apicobasal polarity and promotion of individual cell motility [15]. Hybrid EMT is a multi-faceted process that involves alterations in genomic, epigenomic, morphology, metabolism, proliferative index, immune evasion, and tumour-initiation potential. The hybrid E/M phenotypes may vary both in position of occupancy and regulatory mechanisms [16]. A recent study reported that experimental knockout of zinc finger E-box binding homeobox 1 (ZEB1) gene along with induced expression of either zinc finger protein SNAI1, Slug, or Twist or TGF-β1 exposure may convert E cells to hybrid E/M cells which may be unable to progress into the M state. A stable hybrid E/M state cell population was generated which exhibited high tumourigenicity and lacked plasticity in basal breast cancer cells [17]. Partial EMT poses greater metastatic risk than complete EMT [4] and the combined expression of E and M markers by the hybrid E/M cells confer poor clinical prognosis in cancer [6].

## 3. Markers of Hybrid EMT

Hybrid EMT is generally marked by the co-expression of E and M proteins. Mammospheres produced from E and M sub-clones exhibited the co-expression of CD24^hi^/CD44^hi^ which confirmed the status of hybrid EMT. Other markers included placental-cadherin (P-cad), Slug, and integrin-β 4 (ITGB4). A propagated hallmark of hybrid EMT is collective cell migration where P-cad and Slug are indicated to play pivotal roles [8]. However, hybrid E/M cell populations may not always co-express E and M markers, instead bulk proteomic measurements may also indicate their intermediate nature [18]. Three specific E/M biomarker proteins, regulator of G-protein signaling 16, plasminogen activator inhibitor-2 or SerpinB2 and an integrin α3 are upregulated in partial EMT cells and are not expressed exclusively by E or M cells [4]. Hybrid E/M status was divided into early and late hybrid E/M states according to the expression patterns of the surface markers, such as CD106, CD61, and CD51 [13]. Differential expression of CD104/CD44 determined various E/M states: E (CD104+CD44^lo^), hybrid E/M (CD104+CD44^hi^), and M (CD104-CD44^hi^) cells [17].

## 4. Regulation of Hybrid EMT

Hybrid EMT is governed by phenotypic stability factors (PSFs), transcription factors (TFs), adherens junction proteins, namely E-cad and N-cad, epigenetic regulators, post-translational modifications and TME. Epigenetic regulators (methylation, acetylation, and non-coding RNAs) and post-translational modifications (phosphorylation, ubiquitination, and sumoylation) regulate the expression of various TFs, including SNAI1, Twist1, ZEB1, and ZEB2, which in turn augment EMT and subsequent metastasis. The following section will describe the plethora of varied mechanisms by which hybrid EMT is controlled. Several regulators of hybrid EMT are listed in Table 1.

### 4.1. PSFs Involved in the Maintenance of Hybrid E/M Phenotypes

Several researchers have stated hybrid EMT as a transient state but several factors were found to stabilize or maintain the hybrid state [34]. Cellular clusters expressing Jagged dominated Notch signaling exhibited plasticity with better stability of hybrid E/M phenotypes [35]. In various cancer cells, such as triple negative breast cancer (TNBC) cells SUM159-PT and MDA-MB-231 and 4T1 cells, metabolic preprogramming permitted the malignant cells to perform both glycolysis and oxidative phosphorylation in order to maintain hybrid E/M status against an unfriendly microenvironment [36]. Stochastic simulations have deduced that grainyhead-like transcription factor 2(GRLH2), ovo-like zinc finger (OVOL), ∆Np63α, microRNA (miR) 145/OCT4 play crucial role as PSFs in perpetuation of hybrid E/M phenotypes [37]. Nuclear factor erythroid 2-related factor 2 (Nrf2) has been deciphered as another important PSF for hybrid E/M form, whose constitutive expression upregulated both E-cad and ZEB1 in non-small cell lung carcinoma (NSCLC) and bladder cancer cells [19]. Feedback loops involving E splicing regulatory protein 1, hyaluronic acid synthase 2 and CD44 sustain elevated levels of ZEB1 and ascertain the stability of hybrid E/M phenotypes [38]. Numb, an inhibitor of Notch intercellular signaling, modulate Notch-driven EMT by inhibiting the H1975 lung cancer cells to undergo complete EMT and act as a PSF for maintaining stable hybrid E/M status [39].

### 4.2. TFs Involved with Regulation of Different Stages of EMT

Tumour cells in particular exhibit downregulation of E specific markers, such as the cad1 epithelial cell adhesion molecule (EpCAM), in the early stage of EMT and conserved expression of keratin (Krt) markers—Krt14, Krt5, or Krt8—in hybrid E/M cells which are completely depleted in the late stages of EMT [13]. Transcriptional repression of E-cad (hallmark of malignant tumours) is mediated by EMT-TFs, including ZEB1 and ZEB2, the Snai1 family (Snail, Slug, and Smuc) and basic helix-loop helix factors (Twist and E12/E47) [40]. ZEB1 and ZEB2 play a pivotal role in the positive regulation of EMT phenotypes and aggressiveness of cancer cells [41].

Different transition stages of EMT are associated with E and M specific TFs. Early hybrid EMT stages were initiated with the expression of ZEB1, transformation-related protein 63, Twist 1/2, and LIM/homeobox protein 2 while Smad2 promoted the latter stages of EMT [6]. M markers, including TFs, cad2, vimentin, SNAI1, Twist1/2, and ZEB1/2, were robustly expressed in early hybrid states and were conserved throughout M stages. The late stages of EMT were marked with the prominent expression of cad11, platelet-derived growth factor receptor (PDGFR) A, PDGFRB, fibroblast activation protein, lysyl oxidase like-1, collagen 24A1, matrix metalloproteinase 19 (MMP19), or paired-related homeobox (PRRX)1). EMT is restrained and MET is induced by the OVOL2-mediated downregulation of ZEB1 [42]. The deletion of Twist1 or Snai1 alone was not sufficient to suppress EMT, while the deletion of ZEB1 had a greater impact on the tumour phenotype and metastasis [43]. Transcriptional activation of Snai1 by Wilms’ tumour transcription factor (WT1) prevented repression of E-cad and conferred the hybrid E/M state in renal cancer [20]. Administration of Src kinase inhibitors in ovarian carcinoma cells with hybrid E/M phenotypes, caused regain of E-cad, suppressed Snai1 and Snai2 levels, while ZEB1, ZEB2, and Twist1 levels remained unaltered and contributed towards stabilization of hybrid E/M phenotypes [44]. WT1 evoked the hybrid E/M state in RCC by coherently upregulating SNAI1 and maintaining E-cad expression [20]. Sustained expression of TF, ∆Np63α or suppressing the TGF-β/Smad2 pathway increased the transition towards the early hybrid state than full EMT [13]. In normal Madin–Darby canine kidney (MDCK) cells, YBX1 overexpression induced partial EMT, by increasing several EMT TFs that can regulate tumour angiogenesis and anchorage-independent growth [21]. TGF-β induced PRR homeobox 2 transcription factor upregulation in human breast cells, enhanced migration, and tumourigenesis [22].

### 4.3. Epigenetic Regulators of Hybrid EMT-TFs

Epigenetic regulation of hybrid EMT may be mediated by methylation, acetylation and non-coding RNAs. Chromatin modification by histone deacetylases and/or DNA methyl transferases (DNMTs) facilitate plasticity, aid transformation into hybrid E/M phenotypes, cause transcriptional repression of E genes, stabilise heterochromatin modification and facilitate the recruitment of DNMTs. In partial EMT, the repressed promoters have enriched tri-methylation at the 27th lysine residue of the histone H3 protein (H3K27me3) and impoverished tri-methylation at the 4th lysine residue of the histone H3 (H3K4me3). On the contrary, the activated promoters feature elevated levels of H3K4me3 and acetylation at the 27th lysine residue of the histone H3 protein (H3K27ac). Repressed enhancer regions also harbour mono-methylation at the 4th lysine residue of the histone H3 protein (H3K4me1) with absence/presence of H3K27me3, whereas activated enhancers are marked by H3K4me1 with pronounced H3K27ac [8] GRHL2 was observed to be associated with the epigenetic control of hybrid EMT [45]. The high-mobility group AT-hook 2 was also reported to govern the E–M plasticity and aggravation of intermediate EMT and M state [46]. TF ∆Np63α was associated with upregulation of Slug and Axl (member of the TYRO3-AXL-MER family of receptor tyrosine kinases), downregulation of ZEB1/2 by miR-205 and propulsion towards hybrid EMT in breast cancer [47]. A double negative feedback loop between the miR200 family and ZEB1 was proposed to regulate the E/M transition states [48]. A recently reported model known as the “ceRNA theory” and the nonsense-mediated mRNA decay pathway may be involved in EMT induction through multiple gene-expression networks, including EMT-TFs, in cancer cells [49]. An oncomiR, miR-151a, induced partial EMT and migration in NSCLC cells [23]. The upregulation of long noncoding RNA (lncRNA) myocardin-induced smooth muscle lncRNA, inducer of differentiation (MYOSLID) was associated with the modulation of partial EMT, leading to metastasis and poor prognosis in head and neck squamous cell carcinoma (HNSCC) [24]. Overexpression of lncRNA HOX Transcript Antisense Intergenic RNA (HOTAIR), maintained hybrid EMT phenotypes and induced migration in HCC cells [25].

### 4.4. Post-Translation Modifications Control EMT-TFs and Adherens Junction Proteins

Additionally, post-translational modifications, such as phosphorylation, ubiquitination and sumoylation, are largely known to regulate EMT-TFs [4]. The inhibition of glycogen synthase kinase 3β-mediated phosphorylation and subsequent ubiquitination of SNAI1 induced EMT. The subcellular localization of SNAI1 is regulated by kinases, such as pyruvate dehydrogenase kinase 1, large tumour suppressor kinase 2, and serine/threonine-protein kinase PAK 1, mediated phosphorylation. Apart from phosphorylation, SNAI1 is also controlled by lysine acetylation [8].

The adherens junction proteins, such as E-cad and N-cad, are dominant players of EMT. Post-translation modifications, including serine or threonine phosphorylation, tyrosine phosphorylation, GTPase binding, lipid binding, and proteolysis of the cadhesome proteins, mediate their interaction and activity in adherens junction dynamics [50]. Insulin-like growth factor-II induced the loss of E-cad by novel sumoylation in preclinical models of hepatitis B virus-induced hepatocellular carcinoma [51].

In short, the multiple regulatory networks control EMT-TFs, modify expressions of EMT marker proteins at the transcriptional, post-transcriptional, post-translational and epigenetic levels, and control EMT and aggressiveness in co-operation with additional factors secreted from cells and/or the cellular components in the TME.

### 4.5. TME Influences Hybrid EMT

TME influences both the transformation of the E-to-M form by EMT and the reversion of the M-to-E form by MET at distant metastatic sites [52]. During EMT, the components of the stromal tissue are altered by a marked increase of infiltrating monocytes, macrophages, blood and lymphatic vessels. TME components such as cancer-associated fibroblasts (CAFs), myofibroblasts and tumour-associated macrophages (TAMs), act in a concert and promote tumour development. Though the influence of TME in regulating the hybrid is not well elucidated, some reports have indicated that CAFs, TAMs and stromal components may govern intermediate stages of EMT [53]. Reactive stroma of TME, enriched with FGFs, HGF and tumour necrosis factor α act in association with TGF-β to trigger EMT in cancer cells and facilitate collective migration at invasive front of the multicellular clusters [3]. CAFs inflict tumour proliferation, invasion and metastasis. Collective cancer cell migration always occurs within tracks (generated by protease/force mediated matrix remodelling) in the ECM behind the leading CAF [3]. Single cell migration is mediated by anterior CAFs which secrete membrane type 1 metalloprotease and create microtracks. These microtracks pave the way for single cell migration followed by collective cell migration accompanied with large-scale degradation of lateral ECM interfaces [54]. Since cells exhibiting hybrid E/M characters localize at the junction between tumour hive and tumour stroma, a heterogeneous adherens junction which expressed E-cad on the cancer cell membrane and N-cad on the CAF membrane was propagated to promote fibroblast-led collective cancer cell migration [55]. The activation of coagulation (fibrin/platelets network) surrounding tumour cells shields CTCs against natural killer (NK) surveillance by inducing TGF-β-mediated suppression of C-type, lectin-like, and type II transmembrane glycoprotein NKG2D or by expressing platelet MHC-class I molecules which camouflage CTCs from NK cells [3]. CTCs interact with neutrophil clusters through secretion of chemokines (granulocyte colony stimulating factor, chemokine ligand (CXCL) 1, CXCL8, or CXCL5) and varied adhesive receptors (vascular cell adhesion molecule 1, intercellular adhesion molecule, and β1 integrin) which in turn have been related to poor prognosis in breast cancers [3].

### 4.6. Exosomes in Regulation of Hybrid EMT

Emerging evidence has shown that extracellular vesicles, such as exosomes, are also associated with EMT. Exosomes released from tumour cells contains EMT signature as evident from Ras-transformed MDCK (21D1) cell-derived exosomes. Exosomes may induce EMT on recipient cells upon their uptake [21]. 21D1 exosomes contained the transcription/splicing factor and RNA-binding protein nuclease-sensitive element-binding protein 1 (YBX1/YB-1) which induces partial EMT [21]. Slug-mediated partial EMT caused increased exosomal secretions of fibronectin1, collagen type IIα 1 and native FGG which acts as biomarkers for hybrid EMT in HCC cells [33]. The cells undergoing a partial EMT were also found to secrete cytokines and exosomes that enabled recruitment of macrophages and bone-marrow-derived M cells to sustain inflammation and promote fibrogenesis, respectively [56]. A proteomics study revealed that hypoxic tumour cells produced chemokines and immunomodulatory proteins, e.g., macrophage colony stimulating factor-1-rich exosomes that eluded host immunity and enhanced tumour progression. Thus, hypoxia and exosomes together enabled partial EMT and tumour aggressiveness [57]. Though exosomes carry several EMT signature molecules, more elaborate studies are needed to establish a significant regulatory relationship of exosomes with partial EMT.

### 4.7. Mathematical Modeling as a Predictor of Hybrid EMT Regulation

The computational models have predicted that EMT and MET may not be symmetric processes and cells in a clonal subpopulation may get interconverted in multiple forms E, M or hybrid E/M phenotypes [16]. Recent advancements have been made to explore the enhancer logic and gene regulatory networks (GRNs) that control the different EMT states. Tumour-specific active enhancers of E and M tumour cells are rich with TF activator protein 1, E26 transformation-specific, transcriptional enhanced associate domain, Runt-related transcription factor, and nuclear factor-κ-light-chain-enhancer of activated B cells (NF-κB) binding sites, indicating that the similar cores of TFs activate chromatin remodeling during EMT [13,58]. Mathematical modeling has been advocated for prediction of the GRNs that augmented the E, M, and hybrid states. These models also predict the interaction of E and M TFs and miRs may form a mutually inhibitory loop which may either lead to an E or M phenotype depending on the domination of the specific TF/miR e.g., miR34/ snai1 or miR200/Zeb loops. The mutually inhibitory loop may be also promoted to a hybrid E/M phenotype in the absence of any strong inhibition/expression of the loop members. Mathematical modeling elicited NRF2, OVOL2, GRHL2, Numb and Notch-Jagged signaling as stabilizers of hybrid E/M and suppressors of complete EMT [6]. TFs, OVOL2 or GRHL2, aggravated the hybrid EMT state with high tumourigenic potential and poor patient prognosis [6]. Inhibition of OVOL2 or GRHL2 has been observed to retard collective cell migration [59]. The coupling of OVOL with miR-200/ZEB/LIN28/let-7 network was mathematically deduced to control the stemness of the hybrid E/M phenotype [60]. In the presence of TGF–β, Notch1 has been observed to inhibit Notch3 and activate ZEB1, which in turn has been associated with the promotion of hybrid EMT [61]. Numerical simulations have provided insight that the noise and the time of delay act oppositely on the expression of ZEB, which in turn determines the transition of the E/M spectrum [62]. Bioinformatic strategy observed that the nuclear factor of activated T-cells and specificity protein 1 concerted as master regulators for facilitating a hybrid E/M phenotype in non-transformed mammary gland cells and colorectal cancer cells [6]. A recent transcriptomics based scoring metrics was proposed to be effective in differentiating pure hybrid E/M population from the cluster of E and M cells [63]. Altogether, the computational approaches provide systematic tools to speculate the regulatory factors of the E and M transformation spectrum which further need experimental validations.

### 4.8. Miscellaneous Regulators

NEO1 receptor of the deleted in colorectal carcinoma/Frazzled/UNC-40 family stabilizes E adherens junctions, while loss of NEO1 leads to the disruption of zonula adherens. Loss of NEO1 may lead to metastasis by inducing partial EMT in human colorectal cancer cells by maintaining E connections and acquiring more migratory cellular morphology and changes in gene expressions [26]. DEPTOR protein was found to act as an inhibitor of mammalian target of rapamycin (mTOR). Overexpression of DEPTOR promotes proliferation and survival of cancer cells. DEPTOR induces a partial E-to-M transition and metastasis via autocrine TGF-β1 signaling in HCC [27]. Endosulfan is an organochlorine pesticide that is mainly metabolised by cytochrome P450 (CYP) enzymes, CYT2B6 and CYT3A4, in human liver. Endosulfan was found to induce changes in the expression of epithelial characteristics, disrupted the anoikis process, modulated the cytoskeletal architecture and thus appeared to induce a partial EMT-like event that occurred without cell migration in HepG2 cells [22,28]. Autocrine TGF-β2 signaling facilitated the formation of lipid droplets that promoted partial EMT and helped as energy stores during local invasiveness, anoikis resistance, and distant metastasis in different human cancers [29]. In inflammatory breast cancer (IBC) cells, the colony stimulating factor (CSF)-1/ receptor tyrosine kinase of CSF-1 (CSF-1R) axis has a functional role in hybrid E/M phenotype development and metastasis [30]. mTORC1 and mTORC2 were activated in renal cell carcinoma, and existence of hybrid E/M cells indicated partial EMT [31]. Forkhead box protein C2 expression was associated with co-expression of E/M markers in castration resistant prostate cancer (CRPC)s [32]. Several PSFs and other regulators of hybrid EMT are schematically represented in Figure 2.

## 5. Preclinical and Clinical Evidence of Hybrid EMT

Growing evidence of preclinical and clinical studies exhibited an emerging role of hybrid E/M phenotypes as an important factor of tumourigenesis. The hybrid EMT state of tumour cells remains in a different EMT spectrum than those tumour cells in the completed E/M phenotype [34]. The hybrid EMT state may regulate different factors of tumour progression including co-expression of E/M markers, stemness, tumourigenicity and metastasis along with changes in cellular shape and behavior [64].

### 5.1. In Vitro Studies

In IBC cells, CSF and CSF1R signaling network was observed to drive hybrid E/M phenotypes and aggravate metastasis [30]. Co-expression of E/M markers along with stemness were evident with triple-negative IBC cells [65]. CD24+/CD44+ double positive human mammary epithelial (HMLER) cells co-expressed multiple E/M genes, exhibited stemness and enhanced aldehyde dehydrogenase1 (ALDH1) activity [66]. A hybrid E/M population of HMLER cells resulted into high tumourigenicity and stemness but lacked plasticity [17]. In the hTGF-β1-induced EMT of human mammary MCF10A cells, a certain population of cells exhibited hybrid EMT [67]. E/M double positive hybrid lung adenocarcinoma cells exhibited aggravated migration as M feature and high aggregation property as an E characteristic [18]. Erlotinib-resistant NSCLC cells harboured a subpopulation of hybrid E/M cells, which showed a tendency for in vitro spheroid formation but did not exhibit increased cell migration or invasion in comparison to their erlotinib-sensitive parental cells [68]. PC-3/Mc cells, a subpopulation of prostate cancer PC-3 cells, were proposed to be hybrid E/M cells with higher tumour initiation potential and an active self-renewal program [69]. Primary tumour-derived prostate cancer cells also exhibited a hybrid phenotype along with stemness properties [70]. Human pancreatic cancer cells showed collective migration with hybrid EMT under experimental conditions [71].

### 5.2. In Vivo Studies

A serial xenotransplantation of unsorted rhabdomyosarcoma cells in non-obese diabetic/severe combined immunodeficiency (NOD/SCID) γ (NSG) mice induced hybrid E/M phenotypes [72]. Hybrid EMT was detected in primary colorectal cancer cells and also in its patient-derived xenografts (PDXs) model [73]. In the ovarian cancer xenograft model, some subpopulation of cells were found to be in hybrid E/M multipotent phenotype [74]. Human pancreatic cancer stem cells, which exhibited co-expression of E/M markers were able to induce tumour growth in the xenograft model [75]. In the LSL-KrasG12D; p53loxP/+; Pdx1-cre; LSL-Rosa26YFP/YFP (KPCY) mouse model of pancreatic ductal adenocarcinoma (PDAC), cell populations expressed partial EMT [76]. The crossing of the Pb-Cre+/-;PtenL/L;KrasG12D/+ prostate cancer model with vimentin green fluorescence protein (GFP) reporter strain generated the Pb-Cre+/-;PtenL/L;KrasG12D/+;Vim-GFP (CPKV) mice model which exhibited E, E/M and M states [77]. Co-expressions of E/M markers were also evident in prostate cancer xenografts [78]. During the spontaneous EMT of primary skin tumours in the genetic mouse model of skin squamous cell carcinoma (SCC) with conditional KRasG12D expression and p53 deletion in hair follicles, some subpopulations of tumour exhibited hybrid EMT [13].

### 5.3. Clinical Evidence of Partial EMT in Human Cancers

Several pieces of clinical evidence, mostly with CTCs, showed the existence of hybrid E/M phenotypes during EMT. Partial EMT has been observed with CTCs of the human blood in NSCLC, prostate, breast, liver, colorectal, gastric, and nasopharyngeal cancers [6]. CTCs are regarded as biomarkers of cancer prognosis. In case of advanced cancers, CTCs frequently exhibit pronounced hybrid E/M phenotypes and collective migration of cells [8]. Partial EMT features are also evidenced from M gene expression of the stromal cells around the primary tumours [79] of breast, lung and prostate [8]. In HNSCC, the leading edge of primary tumours elicits partial EMT features and helps in determining nodal metastasis. Single cell transcriptomic analysis from a primary tumour of HNSCC patients exhibited hybrid EMT [80]. In colorectal cancer tumour buds, small groups of tumour cells in the stroma are thought to be representative of partial EMT and poor prognosis [81]. Primary prostate cancer cells exhibited a hybrid E/M phenotype from prostate cancer patients [82]. Co-expressions of E/M markers were also evident from metastatic tumour sites of prostate cancer patients [78]. Hybrid expressions of E/M phenotypes were identified in CTC from metastatic NSCLC patients [83], ovarian cancer patients [84] and patients with metastatic CRPC [85]. The expression of hybrid EMT markers was observed in esophageal squamous cell carcinoma (ESCC) samples from primary tumours with paired metastatic lymph nodes (MLNs) [86]. In early breast cancer patients, EpCAM+CTCs express both M and stemness-related genes; the overexpression of these markers may be associated with worse prognosis [87]. Hybrid expressions of E/M phenotypes were identified in CTC from metastatic breast cancer patients [85]. Co-expression of E/M markers both in the primary breast cancer site and metastatic lymph nodes exhibited the worst survival rates among the patients [88]. Evidence showed tumour buds undergoing partial EMT in colorectal cancer patients [89]. Primary human high-grade serous ovarian cancer (HGSOC) tumour exhibited co-expression of E/M markers [90]. Primary human adenocarcinomas and SCC showed evidence of co-expression of E/M markers along with migratory properties [91].

Emerging evidence of hybrid E/M phenotypes promoting tumourigenic properties in preclinical and clinical models is shown in Table 2.

## 6. Impact of Partial EMT

Cells harbouring intermediate EMT not only impart enhanced metastasis but also inflict cancer stemness and resistance to anoikis, chemotherapy and radiotherapy. The hybrid E/M cells are pleiotropic and aggressive in nature since they have characteristics of autocrine and paracrine signaling, cellular and extracellular attachments, and rapid proliferative property [16].

### 6.1. Aggravated Metastasis

Recent evidence suggests that hybrid EMT may be even more effective than a complete EMT, in spreading metastasis and therapy resistance. In comparison to the isolated expression of vimentin alone, the dual expression of M marker vimentin and E markers cytokeratins 8 and 18, promoted invasiveness, metastasis and poor prognosis in breast cancer [16]. Matrix detachment may upregulate 5′-AMP-activated protein kinase mediated Twist expression necessary for hybrid EMT which is frequently evidenced in CTCs [94]. EpCAM+/CD44+ CTCs isolated from breast cancer patients inflicted bone metastasis in immunodeficient NSG mice [95]. Hybrid EMT increased tumour heterogeneity and imparted advantage during metastasis and drug resistance in IBC [96]. Ovarian cancer has been also evidenced to metastasize by collective cellular migration to form secondary lesions [97]. Partial EMT, by virtue of its tumour heterogeneity, promotes macrometastases. Activation of complete EMT in hybrid PC-3/Mc prostate cancer cells increased their invasiveness, but retarded anchorage-independent growth, stemness and metastasis [98]. Partial EMT in head and neck cancer influenced nodal metastasis, lymphovascular invasion and extranodal extension [80]. Cad-6, a target of TGF-β was found to inhibit autophagy and drive hybrid E/M-mediated aggressive metastasis in thyroid cancer [99].

### 6.2. Cancer Stemness

CSCs are characterized by enhanced tumourigenic capability and have the potential of self-renewal and differentiation into different types of tumour cells. CSCs attain hybrid E/M phenotypes with their existing E character and by concurrently gaining an M character that enhances their motility; thus, they act as leaders of metastasis [100]. EMT has been intertwined with cancer stemness and tumour propagating potential. The hybrid EMT state shows greater propensity to acquire CSC properties [64]. The hybrid E/M cells (CD44^hi^/CD24^hi^) exhibited greater capability of forming mammospheres and formed large tumours in nude mice than E (CD44^lo^/CD24^hi^) or M (CD44^hi^/CD24^lo^) forms [101]. Highly metastatic TNBC cells have also been evidenced with ITGB4+ hybrid E/M cells with CSC property [102]. Activation of the Notch-Jagged pathway maintains clusters of CTCs and stemness in IBC [16]. Breast cancer cells harbouring partial EMT have been associated with the expression of ALDH1, and such cells are potent enough to give rise to both CSCs and differentiated cells [103]. In the CPKV mouse model of prostate cancer, hybrid E/M cells, enriched with Lin-/Sca1+/CD49f^hi^ stem/progenitor cells showed a propensity towards spheroid formation [77]. Hybrid E/M clone P4B6 obtained from OPCT-1 prostate cancer cells exhibited pronounced expression of stem cell markers, such as SOX2, CD44, and NANOG, which in turn caused faster and larger aided tumour formation [70]. In ovarian cancer, hybrid E/M phenotypes induced greater stemness than fully E or fully M phenotypes [74]. It has been reported that, in NSCLC, TGF-β1 promoted the expression of CD133+ in hybrid E/M cells (expressing both E-cad and SLUG) which in turn caused conversion of non-stem cells to CSCs [92].

### 6.3. Therapy Resistance

The impact of hybrid EMT in therapy resistance is yet to be deciphered, although several studies have indicated that intermediate stages of EMT are more vulnerable to chemo- and radio-resistance. NSCLC cells harbouring hybrid E/M characteristics were found to be resistant to epithelial growth factor receptor (EGFR) inhibitor, which in turn increased the ZEB1-mediated sphere formation property [68]. The resistance towards EGFR inhibitors by lung cancer cells has been attributed to cellular memory, especially exhibited by metastable cells [16]. Breast cancer patients with elevated levels of hybrid EMT markers, such as ITGB4 [102], GRHL2 [104] and ΔNp63α [47], were resistant to chemotherapy and exhibited poor relapse-free survival. In ovarian cancer, miR-200c regulates the expression of class III βtubulin and partial EMT, thereby contributing to chemotherapy resistance [16]. A subline of BxPC-3, pancreatic cancer cells with dual expression of ZEB1 and the E-cad characteristic of hybrid EMT exhibited gemcitabine resistance [105]. Colorectal cancer cells, with partial EMT co-expressed ZEB1, ZEB2, Twist and E-cad and elicited resistance against 5-fluorouracil [106].

## 7. Conclusions

Emerging evidence indicates that, instead of complete EMT, hybrid EMT plays a pivotal role in the promotion of cellular plasticity, collective migration, cancer stemness, drug resistance, and metastasis. Despite the recent advancements and in-depth knowledge of the TFs regulated EMT and MET mechanisms, some important areas are yet to be elucidated. The markers of hybrid EMT, specific regulatory molecules involved with the stabilization of the hybrid E/M phenotype and those related with the transition from the hybrid EMT to complete EMT or MET remain unsolved. Computational and bioinformatics approaches may be excellent for identifying new factors or combinations of regulatory elements that govern the different EMT transition states. The EMT scoring metric was shown to differentiate between pure hybrid E/M cells from mixtures of E/M subpopulations by utilizing computational method random circuit perturbation mediated gene expression data [11]. However, careful experimental designs are needed to validate these predictions. However, transcriptional regulation of EMT is not absolute, as the post-transcriptional regulation of hybrid E/M phenotype and EMT associated migration and invasion by Ras has been recently reported [107].

Therapeutic intervention against hybrid E/M cells may evolve as a rational strategy against metastasis and drug resistance. Thapsigargin, a drug which retards lysosomal function and activates an unfolded protein response in the endoplasmic reticulum, acted against hybrid E/M CSCs [CD44^hi^/EpCAM^lo^/CD24^hi^] in CA1 and LM cells [108]. Simvastatin has been recently reported to target cellular plasticity and retard ovarian cancer metastasis as well as stemness by the downregulation of the Hippo/YAP/RhoA pathway [109]. Immunotherapy may be targeted against the immune checkpoint proteins such as programmed death ligand 1 (PD-L1), which have been frequently found expressed in CTCs of NSCLC and TNBC [3]. Saracatinib (AZD0530), an Src-kinase inhibitor, refurbished E-cad in intermediate M cells (Ncad^hi^/ZEB1^hi^/E-cad^lo^) which in turn inhibited spheroidogenesis in anoikis-resistant preclinical models of ovarian cancer [44]. The advent of the hybrid metabolic phenotype has warranted the need of targeting both glycolysis and oxidative phosphorylation for combating cancer metabolic plasticity [110].

In future, the cutting edge technologies, such as concurrent measurement of transcriptome and proteome at unicellular level-CITE-seq [111] and REAP-seq [112], intravital correlative microscopy [113], bioinformatics, and mathematical modeling, might be indicative of the underpinning mechanisms of E/M plasticity and may provide with better anti-cancer therapeutic interventions.

## Figures and Tables

**Figure 1 biomolecules-10-01561-f001:**
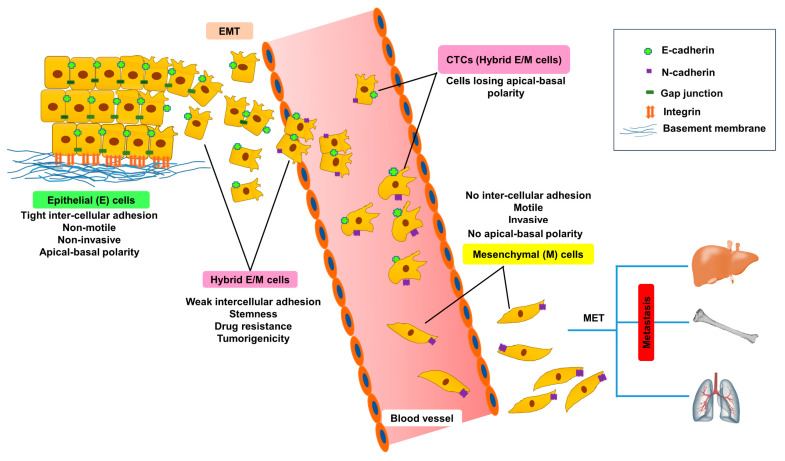
Hybrid E/M cells as a transition state of epithelial mesenchymal transition (EMT). E, M and E/M specific markers and characteristics during different transition states. Loss of cellular adherens of E cells allow them to undergo EMT and are released into the blood circulations as circulating tumour cells (CTCs). CTCs were found to exhibit hybrid E/M phenotypes which can transform into M forms and induce metastasis through M-to-E transition (MET) at distant sites.

**Figure 2 biomolecules-10-01561-f002:**
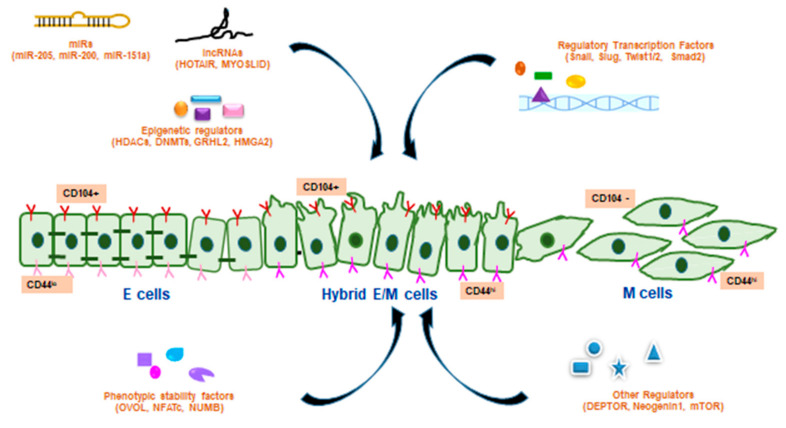
Regulation of hybrid EMT by various factors. The hybrid E/M transition state (CD104+CD44^hi^) between the E (CD104+CD44^lo^) and M (CD104-CD44^hi^) phenotypes are regulated by various factors including micro RNAs (miR), long non-coding RNAs (lncRNAs), other epigenetic regulators, regulatory TFs of hybrid EMT, phenotypic stability factors (PSFs) and other regulators of hybrid EMT.

**Table 1 biomolecules-10-01561-t001:** Regulators of hybrid/partial EMT in various cancer models.

Model	Regulators of Partial EMT	Effect	Mechanisms	Ref.
Human bladder cancer (RT4), NSCLC (H1975) cells	NRF2	Stabilize hybrid EMT	↑E-cad, ↑ZEB1	[19]
Human clear cell RCC (SN12C and ACHN) cells	WT1	Hybrid EMT, tumourigenicity	↑SNAI1, ↑E-cad	[20]
MDCK cells	YBX1		↑SNAI1, ↑Twist, ↑ADAM9, ↑ADAM17, ↑TGF-β1, ↑CSF-1, ↑NGF, ↑VGF, ↑SERPIN E1,	[21]
Human breast (MCF10A) cells	PRRX2	Hybrid EMT, migration, invasion, tumourigenesis, poor survival, aggressiveness	↓CDH1, ↑Vimentin, ↑fibronectin-1, ↑CDH2, ↑SNAI2, ↑ZEB1, ↑TWIST1, ↑tPA	[22]
Human NSCLC (A549, H23) cells	miR-151a	Partial EMT, proliferation, migration	↓E-cad, ↑fibronectin, ↑Slug	[23]
Human HNSCC (Cal27) cell	MYOSLID	Partial EMT, metastasis	↑Slug, ↑PDPN, ↑LAMB3	[24]
Human HCC (SNU-449) cells	HOTAIR	Hybrid EMT, migratory phenotype	↓F-actin stress fibrils, ↑β-catenin, ↑E-cad, ↓Vimentin, ↓c-Met	[25]
Human colorectal cancer (Caco-2) cells	NEO1	Partial EMT, motility, metastasis	↓F-Actin stress-fibres, ↓zonula adherens, ↑MMP1, ↑fibronectin-1, ↑ITGB1	[26]
HCCs from patients	DEPTOR		↑SNAI1, ↑TGF-β1-smad3/smad4 signaling, ↓mTOR	[27]
Human HCC (HepG2) cells	Endosulfan	Hybrid EMT, anoikis resistance	↑SNAI1, ↑Slug, ↑XIAP mRNA, ↑fibronectin,	[28]
Human cervix (SiHa), pharynx (FaDu), colorectal (HCT-116 and HT-29) cancer cells	TGF-β2	Hybrid EMT	↑N-cad, ↑SNAI1, ↑Slug, ↑ZEB1, ↑vimentin, ↓ZO-1, ↓E-cad	[29]
Human inflammatory breast cancer (IBC) (SUM149 and SUM190) cells	CSF-1		↓E-cad, ↑vimentin, ↓Krt18, ↓CLDN1, ↑Twist1, ↑SNAI1, ↓plakoglobin	[30]
RCC from TSC patients	mTORC1 and mTORC2		E-cad+, vimentin+	[31]
CTCs of CRPC patients	FOXC2		E-cad+, N-cad+	[32]
Human HCC (Huh7) cells	Slug	Partial EMT, enhanced motility, chemo-resistance	↑fibronectin-1, ↑Collagen type II alpha 1, ↑FGG	[33]

Abbreviations: ADAM, A disintegrin and metalloproteinases; CLDN1, claudin-1; CRPC, castration-resistant prostate cancer; CSF-1, colony stimulating factor 1; DEPTOR, DEP domain-containing mTOR-interacting protein; FGG, fibrinogen gamma chain; FOXC2- Forkhead box protein C2; HCC, hepatocellular carcinoma; HNSCC, head and neck squamous cell carcinoma; HOTAIR, HOX Transcript Antisense Intergenic RNA; IBC, inflammatory breast cancer; Krt, Keratin; LAMB3, laminin subunit beta 3; MDCK, Madin–Darby canine kidney cell; MMP, matrix metalloproteinase; mTOR, mammalian target of rapamycin; MYOSLID, myocardin-induced smooth muscle lncRNA, inducer of differentiation; NEO1, neogenin1; NGF, nerve growth factor; NRF2, nuclear factor erythroid 2-related factor 2; NSCLC, non-small cell lung carcinoma; PDPN, podoplanins; PRRX2, paired-related homeobox 2 transcription factor; RCC, renal cell carcinoma; tPA, tissue-type plasminogen activator; TSC, Tuberous sclerosis; WT1, Wilm’s tumour transcription factor; XIAP, X-linked inhibitor of apoptosis protein; YBX1, Y box binding protein 1.

**Table 2 biomolecules-10-01561-t002:** Evidence of hybrid E/M phenotypes in various in vitro, in vivo models and clinical studies.

Model	Effect	Mechanism	Ref.
In Vitro Studies			
HMLER cells	Hybrid EMT, stemness	↓Krt 5, ↓Krt 8, ↓pan-cytokeratin, ↓E-cad, ↑vimentin, ↑ZEB1, ↑SNAI1	[17]
Human lung adenocarcinoma (DFCI032, H1650, H1693, HCC827) cells	Hybrid EMT, invasion, migration	E-cad+, Vimentin+, ↑ZEB1, ↑SNAI2, ↑miR-34a	[18]
Human lung adenocarcinoma (H1975) cells	Cell migration, hybrid EMT	↓GRHL2, ↓OVOL2, ↓E-cad (CDH1), ↑ZEB1	[59]
IBC (SUM149, Mary-X and FC-IBC02) cells	Co-expression of E/M phenotype, Stemness	CD44+, ↑TWIST1, ↑E-cad, ↑DSC2, ↑Vimentin	[65]
HMLER cells	Hybrid EMT, plasticity, stemness, mammosphere formation	CD24+/CD44+, ↑ALDH1	[66]
Human mammary epithelial (MCF 10A) cells	Hybrid EMT	E-cad^medium^, vimentin^medium^, SNAI1^hi^, ZEB1^medium^	[67]
Human erlotinib-resistant NSCLC (HCC827) cells	Hybrid EMT, cell migration, spheroid formation	Cad-1+, Vimentin+, ZEB1^hi^	[68]
Human prostate cancer (PC-3/Mc) cells	Hybrid EMT, stemness	CD24+, CD44+	[69]
Primary tumour- derived human prostate cancer (OPCT-1) cells	Co-expression of E/M phenotype	E-cad+, vimentin+, cytokeratin+, fibronectin+, N-cad+, SNAI1+, Slug+	[70]
Human pancreatic cancer (PANC1 and MIAPACA2) cells		↓E-Cad, ↑ZEB1, ↑vimentin	[71]
Human NSCLC (A549, H460), primary NSCLC (LT73) cells		↑CDH1, ↑SNAI2	[92]
**In Vivo Studies**			
Genetic SCC mouse model	Hybrid EMT	Krt 14+, vimentin+	[13]
Primary rhabdomyosarcoma NSTS-11 cells in NSG mice	Hybrid EMT, stemness	↑(ZEB1, MME, LAMC2, or COL3A1), ↓(N-cad, SNAI1, FGF2, AOX1, or ANKRD1), ↑(KRT5, LAMA3, or ANK3), ↓(E-cad, P-cad, KRT14, KRT17, or KRT18)	[72]
Primary human colorectal cancer PDXs in NOG mice	Hybrid EMT, metastasis	E-cad+, ZEB1+	[73]
Primary human ovarian cancer ocv316-X tumour xenograft in SCID-beige mice	Co-expression of E/M markers	E-cad+, Vimentin+	[74]
KPCY mouse model of PDAC		β-catenin+, Claudin-7+, EpCAM+, E-cad+	[76]
Primary prostate cancer CPKV mice model		EpCAM+, Vimentin+	[77]
Prostate cancer DU145 subline in mouse xenografts		E-cad+, ZEB1+	[78]
**Clinical Studies**			
Primary CRC tumour	Hybrid EMT	E-cad+, ZEB1+	[73]
Metastatic tumour sites in prostate cancers patients	Co-expression of E/M markers	E-cad+, ZEB1+	[78]
Primary HNSCC tumours	Hybrid EMT, metastasis	↑Vimentin, ↑integrin α-5, ↑laminins, ↑MMPs	[93]
Primary prostate cancer cells	Co-expression of E/M markers	E-cad+, Vimentin+, Fibronectin+	[82]
CTC from patients with metastatic NSCLC		Vimentin+, Krt+	[83]
CTCs from ovarian cancer patients		EpCAM+, CK5/7+, Muc-1+, N-cad+, Vimentin+, Snai+	[84]
CTCs from patients with metastatic CRPC	Hybrid EMT, stemness	EpCAM+, Cytokeratins+, E-cad+, Vimentin+, N-cad+, O-cad+, CD133+	[85]
CTCs from women with metastatic BT	Hybrid EMT	Cytokeratins+, Vimentin+, N-cad+	[85]
ESCC PT or MLN specimen from ESCC patients		E-cad+, N-cad+, vimentin+	[86]
CTC from early stage breast cancer patients	stemness	TWIST1+, CD44+, ALDH1+, EpCAM+	[87]
Breast cancer samples from primary site and metastatic lymph nodes of breast cancer patients	Co-expression of E/M markers	E-cad+, vimentin+	[88]
Human primary colorectal cancer specimen		Cytokeratin+, vimentin+	[89]
Primary HGSOC tumour		E-cad+, vimentin+	[90]
Primary AC, SCC tumours		Vimentin+/cytokeratin+, E-cad+/N-cad+	[91]

Abbreviations: ANKRD1, ankyrin repeat domain-containing protein 1; AOX1, aldehyde oxidase 1; COL3A1, collagen Type III α1 Chain; FGF2, fibroblast growth factor; LAMC2, laminin subunit γ 2; MME, membrane metallo-endopeptidase.

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
