# Peer review of "Emerging Concepts of Hybrid Epithelial-to-Mesenchymal Transition in Cancer Progression"

_biomolecules, 2020, doi:10.3390/biom10111561_

Round 1
Reviewer 1 Report
Dona Sinha, et al broadly summarized the current knowledge about hybrid EMT. This is an interesting topic that will facilitate the researcher to deeply get insight into EMT process. To improve the paper's quality, it will be helpful if adding more information to relevant places and addressing some concerns.
1. How could researchers discriminate or interpret the difference between partial and fully EMT in their studies especially when they manipulate a gene of interest or other factors through those markers? As we already know, EMT is a dynamic and reversible process, and it has been shown that the cells will undergo multiple stages before reaching to complete M stage. Could we define hybrid EMT as a process in a dynamic transition stage from E to M?
2. In the section of Post Translation Modifications, in fact, other than EMT-TFs, the adherens junction proteins, such as E-cadherin, via post-transcriptional regulation is also a feature of EMT initiation in various EMT models. Could you discuss those in this part?
3. Extracellular vesicles including Large EVs and exosomes in TME are glowingly being considered as a key factor in EMT regulation. The authors should cite some papers and discuss their role in hybrid EMT.
4. In the section of Miscellaneous Regulators, the authors should provide more information about Neogenin1, DEPTOR and Endosulfan.
Author Response
The authors of this manuscript express their sincere thanks to the reviewer for the critical assessment of this work. The authors have acted upon the recommendations of the reviewers which have resulted in a significant enhancement in the quality of this manuscript. All modifications incorporated in the manuscript are highlighted in red color font. A “point-by-point” response to each and every comment is outlined below.
General comments:
Dona Sinha, et al broadly summarized the current knowledge about hybrid EMT. This is an interesting topic that will facilitate the researcher to deeply get insight into EMT process. To improve the paper's quality, it will be helpful if adding more information to relevant places and addressing some concerns.
Response:
The authors are highly thankful to the reviewer for the appreciation and suggestions. The concerns indicated by the reviewer were considered carefully and relevant information has been provided where ever necessary.
Specific comments:
Comment 1:
How could researchers discriminate or interpret the difference between partial and fully EMT in their studies especially when they manipulate a gene of interest or other factors through those markers? As we already know, EMT is a dynamic and reversible process, and it has been shown that the cells will undergo multiple stages before reaching to complete M stage. Could we define hybrid EMT as a process in a dynamic transition stage from E to M?
Response:
Thank you for these excellent thought-provoking questions. The hybrid EMT is manifested especially by the cumulative migration of cells instead of single-cell migration. This has been described in details under Section 2 titled “Hybrid E/M and Their Plasticity” (page 3, lines 86-94). The cells showing such traits might have been investigated with specific hybrid E/M markers and subsequently for the gene of interest. Most of the previous reviews have mentioned hybrid EMT as a plastic process, but we may also certainly describe it as a dynamic transition from E to M. This information has been inserted in Section 2 (page 3, lines 86 and 87).
Comment 2:
In the section of Post Translation Modifications, in fact, other than EMT-TFs, the adherens junction proteins, such as E-cadherin, via post-transcriptional regulation is also a feature of EMT initiation in various EMT models. Could you discuss those in this part?
Response:
This is an excellent suggestion. The post-translation modification of E-cadherin has been inserted under the Section 4.4. “Post Translation Modifications Control EMT-TFs and Adherens Junction Proteins (page 6, lines 220-225).
Comment 3:
Extracellular vesicles including Large EVs and exosomes in TME are glowingly being considered as a key factor in EMT regulation. The authors should cite some papers and discuss their role in hybrid EMT.
Response:
We believe the reviewer had made a terrific point. The role of EVs and exosomes in the regulation of EMT has been widely studied but their regulatory role during hybrid EMT is limited. However, we have discussed this topic under a new Section 4.6. “Exosomes in Regulation of Hybrid EMT; (page 7, lines 258-273).
Comment 4:
In the section of Miscellaneous Regulators, the authors should provide more information about Neogenin1, DEPTOR, and Endosulfan.
Response:
We are in absolute agreement with the reviewer. Relevant information about Neogenin1, DEPTOR, and Endosulfan has been incorporated under Section 4.8. “Miscellaneous Regulators” (page 8, lines 305-316).
Additionally,
- The reference list has been modified as we have added several new references. Special attention is given to conform to the order of references and bibliographic style of the journal.
- The entire manuscript has been thoroughly checked and edited to ensure uniform style, organization, and quality.
Reviewer 2 Report
This manuscript reviews recent progress in the field of EMT during cancer progression.
- Abbreviations should be defined at first mention and used consistently thereafter (line 160-162).
- what is “1 cad1” in line 156
- use the same terminology throughout the document, for example ZEB-1 or ZEB1/deltaEF1, SIP1 or ZEB2 etc
- Partial EMT generates hybrid E/M phenotypes and is characterized into intermediate states. Thus, the term of “Hybrid EMT” in title and main text might cause some confusion for the readers.
Author Response
The authors of this manuscript express their sincere thanks to the reviewer for the critical assessment of this work. The authors have acted upon the recommendations of the reviewers which have resulted in a significant enhancement in the quality of this manuscript. All modifications incorporated in the manuscript are highlighted in a red color font. A “point-by-point” response to each and every comment is outlined below.
General comments:
This manuscript reviews recent progress in the field of EMT during cancer progression.
Response:
The authors are indebted to the reviewer for the kind appreciation and suggestions.
Specific comments:
Comment 1:
Abbreviations should be defined at first mention and used consistently thereafter (line 160-162).
Response:
All abbreviations have been accompanied by full forms for the first time as suggested.
Comment 2:
What is “1 cad1” in line 156
Response:
It was an inadvertent typographical error which has been corrected to “cad1” (page 5, line 160).
Comment 3:
Use the same terminology throughout the document, for example ZEB-1 or ZEB1/deltaEF1, SIP1 or ZEB2 etc
Response:
The terminologies have been corrected to maintain a uniform presentation throughout the document.
Comment 4:
Partial EMT generates hybrid E/M phenotypes and is characterized into intermediate states. Thus, the term of “Hybrid EMT” in title and main text might cause some confusion for the readers.
Response:
We have tried to focus principally on the hybrid E/M phenotype which is generated by the process of partial or hybrid EMT. This has been clarified both in the abstract (page 1, line 24) and introduction (page 2, lines 65 and 66).
Additionally,
- The reference list has been modified as we have added several new references. Special attention is given to conform to the order of references and bibliographic style of the journal.
- The entire manuscript has been thoroughly checked and edited to ensure uniform style, organization, and quality.
Reviewer 3 Report
Sinha et al. have written is a very comprehensive and easy to read review of the main differences between EMT and hybrid EMT. It is quite well established that hybrid EMT gives the cell more plasticity and contributes to more aggressiveness in cancer, and therefore this review is well on point. I very much enjoyed reading it, although I would like the authors to change a couple of minor things:
- In the figure 1 it looks like during tumor development there are no M cells in the tumor, however they have been extensively reported and should be included in it.
- In the same figure 1, even though authors speak about cells losing apical-basal polarity, the cells in the figure have still this polarity. This needs to be corrected.
- Although it is briefly mentioned in the introduction it would be important to extend a bit more the relation of this program with embryonic development. Since this process was initially discovered from a developmental point of view, it would be necessary to include a small section about it. For this the seminal work of Nieto AM. has been crucial, and therefore her publications should be included in the reference list.
Author Response
The authors of this manuscript express their sincere thanks to the reviewer for the critical assessment of this work. The authors have acted upon the recommendations of the reviewer which have resulted in a significant enhancement in the quality of this manuscript. All modifications incorporated in the manuscript are highlighted in a red color font. A “point-by-point” response to each and every comment is outlined below.
General comments:
Sinha et al. have written is a very comprehensive and easy to read review of the main differences between EMT and hybrid EMT. It is quite well established that hybrid EMT gives the cell more plasticity and contributes to more aggressiveness in cancer, and therefore this review is well on point. I very much enjoyed reading it, although I would like the authors to change a couple of minor things:
Response:
The authors are grateful to the reviewer for the encouraging comments and have tried to work on the suggestions which have definitely improved the quality of the manuscript.
Specific comments:
Comment 1:
In the figure 1 it looks like during tumor development there are no M cells in the tumor, however they have been extensively reported and should be included in it.
Response:
In the figure 1, we have tried to show the transition from epithelial to mesenchymal state with intermediate hybrid cells with their marker expressions and other characteristics. As we have not shown tumor development so M cells were not included. Only the transition of E to M cells have been shown.
Comment 2:
In the same figure 1, even though authors speak about cells losing apical-basal polarity, the cells in the figure have still this polarity. This needs to be corrected.
Response:
We admire the reviewer for this critical comment. We have modified the figure and incorporated the changes in the apical-basal polarity of the cells undergoing EMT as suggested.
Comment 3:
Although it is briefly mentioned in the introduction it would be important to extend a bit more the relation of this program with embryonic development. Since this process was initially discovered from a developmental point of view, it would be necessary to include a small section about it. For this the seminal work of Nieto AM. has been crucial, and therefore her publications should be included in the reference list.
Response:
This is an excellent suggestion and we couldn’t agree more. We have discussed this point with the citation of the suggested reference in our revised manuscript (page 1, lines 34-38).
Additionally,
- The reference list has been modified as we have added several new references. Special attention is given to conform to the order of references and bibliographic style of the journal.
- The entire manuscript has been thoroughly checked and edited to ensure uniform style, organization, and quality.